# Prevalence and associated factors for self-reported symptoms of dry eye among Thai school children during the COVID-19 outbreak

Danai Tonkerdmongkol[○], Teera Poyomtip[ID][○], Chotika Poolsanam[ID], Akarapon Watcharapalakorn[ID], Patarakorn Tawonkasiwattanakun[ID]*

Faculty of Optometry, Ramkhamhaeng University, Bangkok, Thailand

○ These authors contributed equally to this work.
* patarakorn.t@rumail.ru.ac.th

**Data Availability Statement:** All relevant data are within the manuscript and its Supporting Information files.

## Abstract

### Purpose

COVID-19 pandemic caused an increase in digital screen time, which seemed to increase the prevalence of dry eye symptoms among the population with abnormally high digital screen usage hours. However, there are no reports of dry eye symptoms in school children with high digital usage hours. Therefore, the present study aimed to assess the prevalence of dry eye symptoms and evaluate the associated factors among school children aged 12 to 18 years during the COVID-19 outbreak.

### Methods

Multistage cluster sampling was applied, and six sections of online questionnaires were distributed to selected respondents in November 2021. The odds ratio (OR) with confidence intervals (CIs) for the factors was calculated using binary logistic regression. All statistical significance was determined at $p < 0.05$.

### Results

The findings revealed that 62.5% of 603 students showed symptoms of dry eye (DEQ-5 score $\geq$ 6). Significant associated factors included being female (adjusted OR (aOR) 1.54; 95% CIs 1.05–2.25), higher-grade student (aOR 1.77; 95% CIs 1.23–2.57), digital screen time use (6 to < 12 hours: aOR 2.00; 95% CIs 1.12–3.57, $\geq$12 hours: aOR 2.54; 95% CIs 1.39–4.76), and perceived stress (aOR 1.12; 95% CIs 1.08–1.16). The Thai-Perceived Stress Scale-10 scores were positively correlated with the scores on the 5-item dry eye questionnaire (Spearman's r = 0.38, $p$-value < 0.01).

### Conclusion

A high prevalence of dry eye symptoms might be common among school children during the COVID-19 outbreak. Significant risk factors include being female, being a higher-grade level student, prolonged use of digital screens, and perceived stress. However, contact lens use,

**Funding:** This research was funded by Faculty of Optometry, Ramkhamhaeng University. The funder has no role in the study design, data collection and analysis, decision to publish, or preparation of the manuscript.

**Competing interests:** The authors have declared that no competing interests exist.

smoking, and the most common digital device usage patterns were not found to be contributing factors.

## Introduction

The severe acute respiratory syndrome coronavirus 2 (SARS-CoV-2) virus that caused the coronavirus disease (COVID-19) pandemic has resulted in public health crises and disrupted normal daily activities [1]. Education is one area that has been adapted to accommodate rapidly changing government policies. According to UNESCO, public health measures to combat the infection have impacted around 365 million students globally. The schools in Asia were shuttered for roughly 40 weeks [2] and replaced with online learning to continue learning. These modifications to online education have resulted in increased exposure time to the digital devices used by school children [3], which has a negative impact on ocular surface [4, 5] and leads to dry eye [6].

The prevalence of dry eye symptoms has increased among school children over the past two decades [7–9], possibly due to a rise in screen exposure when interacting with the internet and social media. Prolonged screen use can cause dry eye symptoms by decreasing the blink rate and drying the ocular surface through evaporation [10, 11]. The resulting symptoms, including dryness, irritation, pain, eye fatigue, and deteriorated vision [12] can have a negative impact on school-related activities. Moreover, the Tear Film and Ocular Surface Society (TFOS) released the Dry Eye Workshop II (DEWS II) Epidemiology report, also pointing out the importance of performing studies in school children or younger populations, which may have a relatively high prevalence to evaluate the potential risk factors, especially the use of digital screens [13].

Thus far, the prevalence and risk factors of dry eye symptoms in school children have been reported in China [9], Japan [14], and Mexico [15], whereas studies in Thailand have been focused primarily on the elderly and university students [16–21]. Among these studies [9, 14, 15], the results of well-known risk factors, such as gender and contact lens use, were inconsistent. In addition, smoking and perceived stress as contributing factors of dry eye symptoms remain under investigation. Moreover, the study in school children was conducted before the COVID-19 pandemic, so the result may not accurately reflect the situation digital screen time induced dry eye, which has increased as a result of lifestyle changes [22]. Consequently, it is still necessary to assess the prevalence and identify risk factors for dry eye symptoms among school children. In this regard, the present study aimed to assess the prevalence of dry eye symptoms and digital screen time during the COVID-19 pandemic among students in grades 7 to 12 attending urban secondary schools.

## Methods

This constituted an online cross-sectional study conducted among grade 7 to 12 students in Samutprakarn Province, Thailand. The Research Ethics Review Committee for Research Involving Human Research Participants Group I at Chulalongkorn University (COA No. 208/201) approved the study protocol on October 7, 2021. Data collection was spanned from 1st to 30th November 2021. In section 1 of the online questionnaires, informed consent (the click-if-you-agree type) was obtained from the parents and respondents.

## Population and sampling method

The population consisted of respondents studying in Samutprakarn Province, and the expected sample size was calculated as follows: $\frac{n = z_{\alpha/2}{}^2 Np(1-p)}{d^2(N-1) + z_{\alpha/2}^2 p(1-p)}$, where N is the population size, 50,362 [23], $z_{\alpha/2}^2$ is the standard normal deviation corresponding to a 95% confidence level, set at 1.96, and p is the prevalence of dry eye, assumed to be approximately 50% [24]. The margin of error, denoted by was set at 5%. The estimated sample size (n) was 382 individuals. Due to the lack of response rate data for online surveys in secondary schools, a response rate of approximately 30% was assumed for the web-based study [25]. Additions of 885 were added to the sample size. Consequently, total of 1,266 participants were recruited for this study.

This study examined 25 secondary schools in Samutprakarn Province, clustered into four Secondary Educational Service Areas. Random sampling was applied to randomly select schools from these Secondary Educational Service Areas. Four schools, one from each Educational Service Area, were selected as the study sites. Further, each school was divided into six Grade levels (Grades 7 to 12). Next, two classrooms were selected at randomly from each grade at individual schools, and students were randomly selected using their student numbers from these classrooms. The exact number of samples was chosen at random based on the number of students in each school. When the total number of samples was collected from each school, it was used to calculate the number of available students in each grade level. Subsequently, the research tool was distributed to the selected students via Google classroom or the Line application group for the selected classrooms.

The study included students in grades 7 through 12 between the ages of 12 and 18 years who voluntarily consented to participate. Students who were blind (no light perception) in both eyes, had an eye infection during the survey, had an underlying disease, such as allergy, diabetes, or autoimmune disease, or used antihistamines on a regular basis were excluded from the study.

## Translation of 5-item dry eye questionnaire (DEQ-5)

The original version of the DEQ-5 [26] was independently translated into the Thai language by a native Thai-optometrists and a native Thai speaker without a medical or clinical background. Next, these two translated versions were combined by the third independent translator, followed by a back translation of the combined version by an individual fluent in the original language. To curate the Thai-DEQ-5, the back-translated version was compared to the original version, and the research team consulted an expert ophthalmologist regarding discrepancies between the versions. The reliability of the Thai-DEQ-5 was examined with 30 students, yielding a Cronbach's alpha coefficient of 0.847.

## Online questionnaire

This study used a Thai online structured questionnaire comprising six sections and a total of 23 questions. **Section 1.** *Brief information, consent, and exclusion questions*: The first page of the online questionnaire contained brief information related to the meaning and symptoms of dry eye (definition of eye discomfort), followed by the consent and exclusion questions. **Section 2.** *Personal factors*: This section contained a total of two questions, including gender and academic year. **Section 3.** *Behavioral factors*: This section consisted of two questions regarding contact lens wear and smoking. **Section 4.** *Digital device factors*: This section consisted of two questions, including the most commonly used digital device types and digital screen time (the number of hours daily). **Section 5.** *Symptoms of Dry eye*: This section consisted of five questions about dry eye (symptoms/occurrence). The responses to three questions ranged from 0

(never) to 4 (constantly), and the responses to the remaining two questions ranged from 0 (never have it) to 5 (very intense). The possible values ranged from 0 to 22, with a cut-off of ≥6.0 indicating symptoms of dry eye and a DEQ-5 score of ≥12 indicating severe symptoms of dry eye [27]. **Section 6.** *Stress*: This section often consisted of questions that assessed perceived stress. The Thai-Perceived Stress Scale-10 (T-PSS-10) [28] was utilized to assess perceived stress levels. This instrument was a 5-point Likert scale with six negative items ranging from 0 (never) to 4 (the most frequent) and four positive items ranging from 4 (never) to 0 (the most frequent). The possible values ranged from 0 to 40, with higher numbers indicating more stress [29]. Related studies showed that Cronbach's alpha coefficient was approximately 0.71 to 0.87 among Thai secondary school students [30–32], and the present study showed it to be 0.82.

## Data analysis

SPSS version 18.0 was utilized for all statistical analyses. Descriptive statistics such as percentage, mean, and standard deviation (SD) were applied for data characterization. The odds ratios (ORs) with 95% confidence intervals (95% CI) were calculated using logistic regression to identify the association, whereas multivariable logistic regression was used to investigate all significant aspects in the univariate analysis. Spearman's correlation was used to establish the relationship between the DEQ-5 and T-PSS-10 scores. The Mann-Whitney U test was used to compare the continuous variables between the two groups, whereas the chi-square test was applied for non-continuous variable comparison. All statistical significance was based on a *p*-value of <0.05. The scatter density plot and box plot were reconstructed from Scimago Graphica.

## Results

### Respondents' demographics and prevalence of dry eye

657 students responded to the questionnaire (51.9% response rate). Of these, 621 respondents consented to participate in this study. However, 18 (2.9%) respondents were excluded due to an eye infection or the constant use of antihistamines. Thus, 603 respondents were enrolled in this study, including 182 (30.2%) males and 421 (69.8%) females. 356 (59.0%) of all respondents were students in grades 7 through 9, while 247 (41.0%) respondents were students in grades 10 through 12. Among the respondents, only 25 (4.1%) respondents wore contact lenses. A small fraction (4.1%) of the respondents had experience with smoking. The majority of respondents (86.7%) self-reported that their primary digital devices were smartphones and tablets. More than one-half of the 603 (54.2%) respondents reported daily digital screen time between 6 and 12 hours. The overall mean ± SD of the Perceived Stress Scale was 19.0 ± 5.3 (Minimum = 4, Maximum = 39). All characteristics of the study respondents are summarized in Table 1.

In this study, the overall prevalence of dry eye symptoms was 62.5%, and the overall mean (SD) DEQ-5 score was 7.2 (4.0). The general demographics and the comparison between the healthy respondents and those with symptoms of dry eye are shown in Table 2. The prevalence of dry eye symptoms was significantly higher in females (67.2%) than in males (51.6%). Students in grades 10 to 12 (71.3%) had symptoms of dry eye more frequently than students in grades 7 to 9 (56.2%). Interestingly, the prevalence of dry eye symptoms among contact lens wearers was 80.0%. The respondents with more time on digital screens exhibited an increased prevalence (39.1 to 71.2%).

**Table 1. General demographics of respondents enrolled in this study (N = 603).**

| Variables | N (%) |
|---|---|
| **Gender** | |
| Male | 182 (30.2) |
| Female | 421 (69.8) |
| **Academic year** | |
| Grade 7 student | 119 (19.7) |
| Grade 8 student | 121 (20.1) |
| Grade 9 student | 116 (19.2) |
| Grade 10 student | 85 (14.1) |
| Grade 11 student | 79 (13.1) |
| Grade 12 student | 83 (13.8) |
| **Contact lens wear** | |
| No | 578 (95.9) |
| Yes | 25 (4.1) |
| **Smoking** | |
| No | 578 (95.9) |
| Ex-smoker | 25 (4.1) |
| **Major types of digital devices use** | |
| Hand-held devices (Smartphone and Tablet) | 523 (86.7) |
| Computer (Laptop or desktop) | 80 (13.3) |
| **Digital screen time (hours used per day)** | |
| <6 hours | 64 (10.6) |
| 6 hours—<12 hours | 327 (54.2) |
| ≥12 hours | 121 (35.2) |

## Correlation of DEQ-5 score with T-PSS-10

Spearman's correlation analysis revealed a significant correlation between the DEQ-5 scores and the T-PSS-10 scores ($r = 0.38$, $p$-value $< 0.01$). The density scatter-plot between the DEQ-5 and T-PSS-10 scores is presented in Fig 1A. Moreover, respondents were classified as having healthy eyes (DEQ-5 score $< 6$), mild-to-moderate dry eye symptoms (DEQ-5 score $\geq 6$ to $<12$), or severe dry eye symptoms (DEQ-5 score $\geq 12$). The mean T-PSS-10 scores increased significantly in the mild-to-moderate and severe dry eye symptoms categories relative to the healthy status ($p$-value $< 0.01$) (Fig 1B).

In the univariate analysis, wearing contact lenses (cOR 2.47; 95% CIs 0.92 to 6.69), smoking (cOR 0.64; 95% CIs 0.29 to 1.42) and major type of digital device use (cOR 1.06; 95% CIs 0.65 to 1.73) were not significantly associated with dry eye. However, being female (cOR 1.92; 95% CIs 1.35 to 2.74), a student in grades 10 to 12 (cOR 1.97; 95% CIs 1.39 to 2.79), digital screen time (6 to <12 hours: OR 2.49; 95% CIs 1.44 to 4.31, ≥12 hours: cOR 3.86; 95% CIs 2.15 to 6.92 when compared to <6 hours), and higher perceived stress score (cOR 1.13; 95% CIs 1.09 to 1.17) were likely to increase the risk of dry eye (Table 3).

All significant factors from the univariate analysis were further analyzed using multivariable binary logistic regression. Females had a 1.54-fold increased risk of symptomatic dry eye (aOR 1.54; 95% CIs 1.05 to 2.25) compared to males. Students in grades 10 to 12 were 1.77 times more likely to suffer from the dry eye than those in grades 7 to 9 (aOR 1.77; 95% CIs 1.23 to 2.57). Respondents with increased digital screen time (6 to <12 hours: aOR 2.00; 95% CIs 1.12 3.57, ≥12 hours: aOR 2.54; 95% CIs 1.39 to 4.76) and higher perceived stress scores (aOR 1.12; 95% CIs 1.08 to 1.16) were more likely to experience dry eye symptoms (Table 3).

**Table 2. General demographics between healthy respondents and those with symptoms of dry eye (N = 603).**

| Variables | Healthy | Dry eye symptoms | p-value |
|---|---|---|---|
| | (n,%) | (n,%) | |
| **Gender** | | | |
| Male | 88(48.4) | 94 (51.6) | <0.01[a] |
| Female | 138 (32.8) | 283 (67.2) | |
| **Academic year** | | | |
| Grade 7–9 | 156 (43.8) | 200 (56.2) | <0.01[a] |
| Grade 10–12 | 70 (28.3) | 177 (71.3) | |
| **Contact lens wear** | | | |
| No | 221 (38.3) | 357 (61.8) | 0.07[a] |
| Yes | 5 (20.0) | 19 (80.0) | |
| **Smoking** | | | |
| No | 214 (37.0) | 364 (63.0) | 0.27[a] |
| Ex-smoker | 12 (48.0) | 13 (52.0) | |
| **Major types of digital devices use** | | | |
| Hand-held device | 197 (37.7) | 326 (62.3) | 0.06[a] |
| Computer (Laptop or desktop) | 29 (36.3) | 51 (63.8) | |
| **Digital screen time (hours used per day)** | | | |
| <6 hours | 39 (60.9) | 25 (39.1) | <0.01[a] |
| 6 hours—<12 hours | 126 (38.5) | 201 (61.5) | |
| ≥12 hours | 61 (28.8) | 151 (71.2) | |
| **Perceived stress** | | | |
| T-PSS-10 (mean ± SD) | 17 ± 5 | 20 ± 5 | <0.01[b] |
| **DEQ-5** | | | |
| DEQ-5 score (mean ± SD) | 3 ± 2 | 10 ± 2 | <0.01[b] |

[a] T-PSS-10: Thai-Perceived Stress Scale-10, DEQ-5: 5-item dry eye questionnaire, SD: Standard deviation, [a]By Pearson's Chi-square test

[b]By Mann-Whitney U test

## Sensitivity analysis

There are no validated questionnaires for dry eye symptoms or cut-off scores currently available for use with children. To increase confidence in the outcomes, a sensitivity analysis using different DEQ-5 cut-off scores ranging from ≥4 to ≥ 8, was performed, which showed that the prevalence ranged from 47.8% to 78.1%. Analyzing the associated factors (sex, grades, digital screen time, and perceived stress) with dry eye produced similar results to the primary findings (Fig 2)

## Discussion

Our results demonstrated a 62.5% prevalence and associated factors of dry eye symptoms during the COVID-19 pandemic, marking the first time school children have been the primary focus (aged 12 to 18 years). In comparison with the survey using dry eye symptom questionnaires in school children pre-COVID-19 pandemic, the present study showed a higher prevalence than studies by *Uchino et al.* [14] (21.6%-2008) and *Zhang et al.*, [9] (23.7%-2010) of dry eye symptoms by using a Women's Health Study questionnaire (WHS questionnaire). However, the prevalence shown in the present study is closer *Garza-Leon et al.*'s [15] study, with 65.3% of respondents in Mexico reporting symptoms of dry eye by using the ocular surface

A

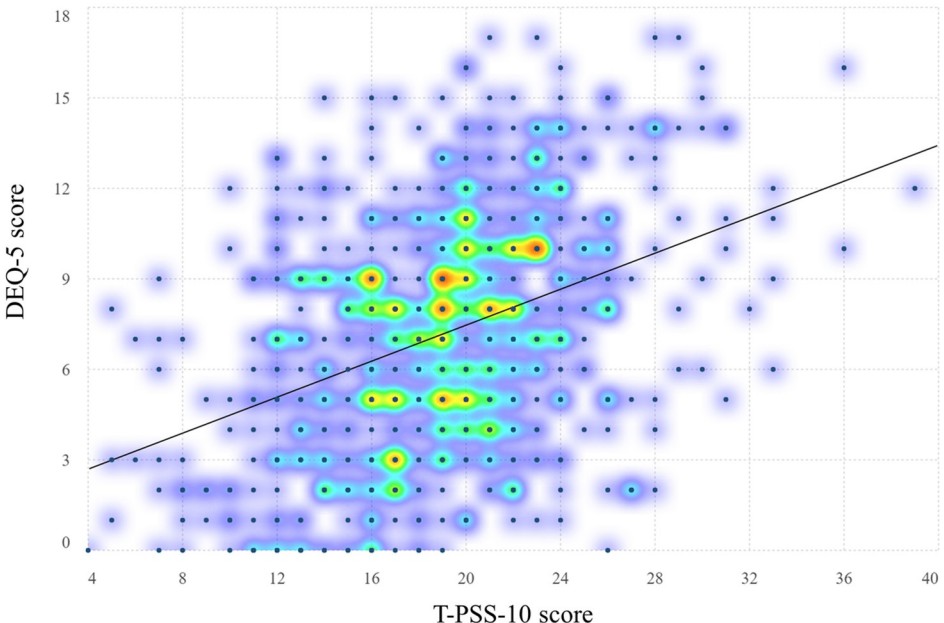

B

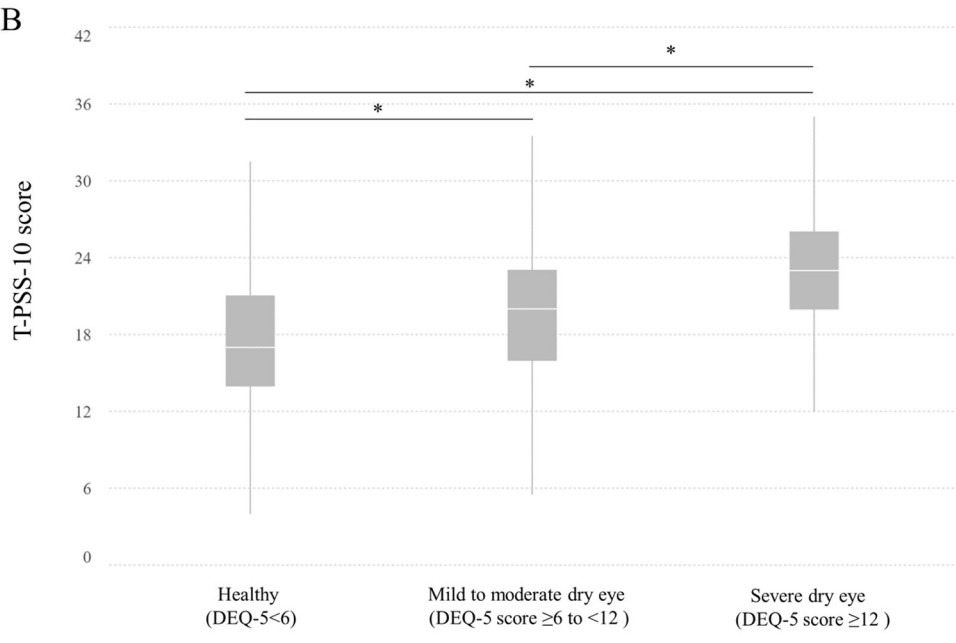

**Fig 1. Distribution of T-PSS-10 and DEQ-5 scores.** (A) Density scatter-plot with a correlation line between the T-PSS-10 and DEQ-5 scores of 603 respondents. Yellow indicates a higher density of points and blue indicates a lower density. (B) Box plot indicates the median T-PSS-10 among healthy, mild-to-moderate, and severe symptoms of dry eye. The asterisk shows statistically significant differences analyzed using the Mann-Whitney U test at $p$-value < 0.01.

**Table 3. Univariate and multivariate logistic regression among study respondents (N = 603).**

| Variables | univariate logistic regression | | multivariate logistic regression | |
|---|---|---|---|---|
| | cOR (95% CI) | p-value | aOR (95% CI) | p-value |
| **Gender** | | | | |
| Male | -Ref- | | -Ref- | |
| Female | 1.92 (1.35–2.74) | <0.01 | 1.54 (1.05–2.25) | 0.03 |
| **Academic year** | | | | |
| Grade 7–9 | -Ref- | | -Ref- | |
| Grade 10–12 | 1.97 (1.39–2.79) | <0.01 | 1.77 (1.23–2.57) | <0.01 |
| **Contact lens wear** | | | | |
| No | -Ref- | | | |
| Yes | 2.47 (0.92–6.69) | 0.07 | - | |
| **Smoking** | | | | |
| No | -Ref- | | | |
| Ex-smoker | 0.64 (0.29–1.42) | 0.27 | - | |
| **Major types of digital devices use** | | | | |
| Hand-held device | -Ref- | | | |
| Computer (Laptop or desktop) | 1.06 (0.65–1.73) | 0.81 | - | |
| **Digital screen time (hours used per day)** | | | | |
| <6 hours | -Ref- | | -Ref- | |
| 6 hours—<12 hours | 2.49 (1.44–4.31) | <0.01 | 2.00 (1.12–3.57) | <0.01 |
| ≥12 hours | 3.86 (2.15–6.92) | <0.01 | 2.54 (1.39–4.76) | <0.01 |
| **Perceived stress** | | | | |
| T-PSS-10 (per score) | 1.13 (1.09–1.17) | <0.01 | 1.12 (1.08–1.16) | <0.01 |

cOR: Crude odds ratios, aOR: Adjusted odds ratio, CI: Confidence interval, T-PSS-10: Thai-Perceived Stress Scale-10, Statistical significance at $p$-value < 0.05.

index disease (OSDI) questionnaire. Contrary, the prevalence shown in the present study is slightly lower than the study conducted by *Lin et al*. [8] during the COVID-19 pandemic, indicating 70.5% prevalence by using the OSDI scores in high school children.

The present findings also revealed a higher prevalence (62.5%) than other studies conducted among university students in Thailand. *Supiyaphun et al*. [17] Estimated 8.15% had symptomatic dry eye using the WHS questionnaire before the COVID-19 outbreak. Similar to the DEQ-5, the WHS questionnaire is short and simple; however, it includes question pertaining to history of the dry eye diagnosis, whereas the DEQ-5 only evaluates the symptoms. Subsequently, *Tawonkasiwattanakun et al*. [20] demonstrated that 10.8% the dry eye symptoms occurred among open university members, students, and staff aged ≤29 years using the McMonnies Questionnaire at the onset of the COVID-19 pandemic (May to June 2020). This questionnaire differs from the others since the questions evaluate dry eye symptoms as well as known risk factors, such as age, sex, and systemic conditions associated with dry eye. Thus, the McMonneies questionnaire may be suitable for clinical screening, and adult populations. As aforementioned, the difference in prevalence may be a result of the growing popularity of online learning and intensive use of digital screens, which can lead to the development of the dry eye [22, 33] due to public health measures assigning the Thai students learning online until December 1, 2021.

In terms of digital screen time, our results were consistent with related reports indicating that the increased use of digital screen time causes dry eye [22, 33–35]. Notably, the relationship between the duration of digital device use and dry eye varied significantly between studies

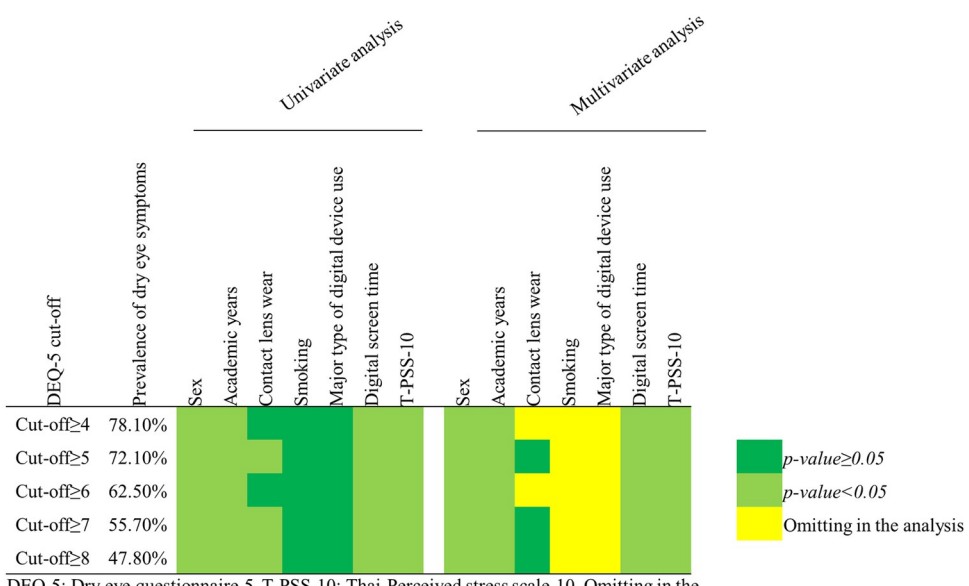

DEQ-5: Dry eye questionnaire-5, T-PSS-10: Thai-Perceived stress scale-10, Omitting in the analysis due to insignificance in univariate analysis

**Fig 2. The prevalence and associated factors of symptomatic dry eye classified using different 5-item dry eye questionnaire (DEQ-5) cut-off scores.** The associated factors were computed by univariate and multivariate logistic regression. The yellow color indicated omissions in the analysis, green color indicated $p$-values $\geq 0.05$ and light green color indicated $p$-values $< 0.05$.

[25, 33, 36]. *Akib et al.'s* study [33] in Indonesia suggested the use of a digital screen for more than three hours triggered symptoms and signs of dry eye, whereas *Muntz et al.* [34] showed that extended screen time use in terms of weekly screen hours was associated with poorer symptomology, elevated blink rates, and proxy tear film stability. These differences may have been caused by additional variables not considered in these studies. *Rossi et al.* [36] found that the number of years and duration of daily breaks from digital screen time varied significantly between groups with and without dry eye.

In addition to screen time, perceived stress may also play a role in the symptomatic dry eye during COVID-19. Previous reports suggested that learning online and being physically inactive since implementing public health measures has been linked to increased perceived stress [37, 38], which has been suggested as a risk factor for symptoms of dry eye in participants aged 15 and older [39].

Although the pathophysiology underlying the relationship is unknown, several hypotheses have been advanced to explain it. First, the relationship may be a vicious cycle in which dry eyes and perceived stress mutually exacerbate one another. The severity of dry eye negatively affects daily life activities [40] and is likely to increase perceived stress [41], as well as induce cortical production and promote pain perception [42], resulting in sensitivity to dry eye symptoms. Such conditions were supported by our results, as shown in Fig 1, in which the DEQ-5 scores were positively correlated with perceived stress, and the severe dry eye symptom group showed the highest T-PSS-10 score. Secondly, perceived stress quite possibly induces pro-inflammatory and inflammatory cytokine secretion, which disrupts tear film homoeostasis and causes ocular surface inflammation [13, 43, 44]. Notwithstanding, our findings suggest that perceived stress is positively associated with dry eye symptoms, which might be prevalent among school children experiencing perceived stress.

The present study had a few limitations. First, the study relied solely on self-assessment to evaluate dry eye. The clinical examination may be necessary to confirm the reported cases.

However, DEQ-5 is one suggested method to evaluate dry eye symptoms by the Dry Eye Workshop II [45]. Second, none of the validated symptom questionnaires currently available address dry eye in school children. In addition, the Thai-DEQ-5 has not been validated with clinical signs, and a cut-off score $\geq 6$ is commonly used in adults older than 18 years. Though, the DEQ-5 has been recommended for use to assess symptoms of dry eye in children [46]. Moreover, sensitivity analysis revealed similar results suggesting the robustness of the outcomes. Thirdly, our study sites were selected from urban schools. Consequently, the results do not represent school children residing in other locations i.e., rural areas, and the study population does not represent the national population. Lastly, contact lens wear and smoking did not show a significant relationship with the dry eye; this could be due to the small sample size of the concerned groups. Regarding smoking, the formal education study environment might contain a small number of smokers. Further research should utilize both clinical examinations and validated questionnaires to confirm the results, and a multicenter study is required to estimate the prevalence of dry eye symptoms and their associated factors in the general population.

## Conclusion

To conclude, the present study revealed for the first time in Thailand that the prevalence of symptomatic dry eye among school children was high during public health measures, i.e., online education and home isolation, in response to the COVID-19 pandemic. Additionally, the prolonged use of a digital screen, increased perceived stress, older age, and female gender was associated with dry eye symptoms. This information should be shared with stakeholders such as parents, school personnel, and teachers in order to develop preventative measures. Guidelines or campaigns for online education, such as taking breaks during online learning, and encouraging physical activity at home, should be considered.

## Supporting information

**S1 Table. Shows data for this study.**
(SAV)

## Acknowledgments

The authors are appreciative of the teachers who approved and distributed the online questionnaire to their students. Additionally, we would like to thank every student who participated in this study. The authors appreciate Professor Dr. Nahathai Wongpakaran for providing permission to utilize the Thai-PSS-10 scale. The authors would like to thank Enago (www.enago.com) for the English language review.

## Author Contributions

**Conceptualization:** Patarakorn Tawonkasiwattanakun.

**Data curation:** Danai Tonkerdmongkol, Teera Poyomtip, Chotika Poolsanam, Akarapon Watcharapalakorn.

**Formal analysis:** Teera Poyomtip.

**Investigation:** Patarakorn Tawonkasiwattanakun.

**Methodology:** Teera Poyomtip, Akarapon Watcharapalakorn, Patarakorn Tawonkasiwattanakun.

**Project administration:** Teera Poyomtip, Patarakorn Tawonkasiwattanakun.

**Supervision:** Danai Tonkerdmongkol, Patarakorn Tawonkasiwattanakun.

**Visualization:** Teera Poyomtip.

**Writing – original draft:** Teera Poyomtip, Patarakorn Tawonkasiwattanakun.

**Writing – review & editing:** Patarakorn Tawonkasiwattanakun.

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
