## [Decision Letter · Decision Letter 0]

6 Dec 2022

PONE-D-22-22320Prevalence and associated factors for self-reported symptoms of dry eye among Thai schoolchildren during COVID-19 outbreakPLOS ONE

Dear Dr. Tawonkasiwattanakun,

Thank you for submitting your manuscript to PLOS ONE. After careful consideration, we feel that it has merit but does not fully meet PLOS ONE’s publication criteria as it currently stands. Therefore, we invite you to submit a revised version of the manuscript that addresses the points raised during the review process.

We look forward to receiving your revised manuscript.

Kind regards,

Michael Mimouni

Academic Editor

PLOS ONE

Journal Requirements:

a) Did participants provide their written or verbal informed consent to participate in this study?

3. You indicated that you had ethical approval for your study. In your Methods section, please ensure you have also stated whether you obtained consent from parents or guardians of the minors (participants below the age of 18 years) included in the study or whether the research ethics committee or IRB specifically waived the need for their consent.

4. We suggest you thoroughly copyedit your manuscript for language usage, spelling, and grammar. If you do not know anyone who can help you do this, you may wish to consider employing a professional scientific editing service. 

This work was financially supported by the 2021 Fund from the Faculty of Optometry, Ramkhamhaeng University.

However, funding information should not appear in the Acknowledgments section or other areas of your manuscript. We will only publish funding information present in the Funding Statement section of the online submission form. 

This research was funded by Faculty of Optometry, Ramkhamhaeng University. The funder has no role in the study design, data collection and analysis, decision to publish, or preparation of the manuscript.

Reviewers' comments:

Reviewer's Responses to Questions

**Comments to the Author**

1. Is the manuscript technically sound, and do the data support the conclusions?

Reviewer #1: Partly

Reviewer #2: Yes

2. Has the statistical analysis been performed appropriately and rigorously? 

Reviewer #1: I Don't Know

Reviewer #2: Yes

3. Have the authors made all data underlying the findings in their manuscript fully available?

Reviewer #1: Yes

Reviewer #2: Yes

4. Is the manuscript presented in an intelligible fashion and written in standard English?

Reviewer #1: No

Reviewer #2: Yes

5. Review Comments to the Author

Reviewer #1: The article addresses an important and relevant subject that might interest the audience of the journal. The study answers the Dry Eye Workshop II (DEWS II) Epidemiology report of the Tear Film and Ocular Surface Society (TFOS), in the context of increased exposure to digital media during the covid-19 pandemic and among younger children. However, the study quotes other studies that examined similar age groups and each contributing factor for dry eye. The novelty of the current research is not well established, besides the context of the accelerated exposure to digital devices during the covid-19 pandemic.

The Literature review sets the stage for the study questions, supported by updated and relevant studies. The literature review is based on relevant studies about self-reported symptoms and clinical examinations of young adults and youth, related to exposure-time to digital media. The authors should re-examine that the review relates to all variables examined, e.g., it lacks information regarding gender differences from previous studies.

The abstract points out that smoking and contact lens wear were not predisposing factors for dry eye, however the relevant groups were too small to have a significant statistical conclusion (not mentioned in the limitations section).

The researchers' assumption of dry eye prevalence greater than 50% should be further established, since all the formula calculations are based on this rate. This rate seems high according to the best of my knowledge of the medical literature, especially among younger populations.

The literature review leads to the study aims and the clear research questions but lacks a more specific hypothesis.

Method

the sampling process and the research population are well described. The research tools are well described, revealing quite high reliability, although they were originally established for adults. Authors should add information regarding translation validation.

Results

The authors provided rich information accompanied by detailed tables and plots, demonstrating the higher prevalence among girls, higher grades, prolonged exposure to screens, and higher perceived stress.

The authors add information regarding the relations between dry eye symptoms and perceived stress that was prevalent during the pandemic. In addition, they identified the daily cutoff exposure time to digital media devices, analyzed by odds ratio.

The discussion deals with the findings in relation to previous studies in the literature. The authors demonstrate a critical approach in the study limitations section, especially the lack of clinical evidence to dry eye. It is very relevant in teenagers' populations, which often demonstrate confounding results, especially regarding to self-reports of smoking in the formal educational system context.

Conclusions

The conclusion repeats the study findings and lack the bottom-line importance and significant contribution of the study, as well as lessons learned and recommendations for the future.

Reviewer #2: Thank you for the Manuscript. I really liked it. I think that DED is not studied enough in this group age and in this set up (covid19 time). please review my remarks I mention and then it's ready to be published.

Thank you for allowing me to review this interesting paper.

Research article - is suitable for this publish.

Remarks

abstract

1. No remarks.

Intro

2. You mention “causing a negative impact on ocular surface” but the reference you bring showed only change in OSDI and not the other Ocular surface findings which showed no correlation.

3. Line 69-70 you write about increased screen exposure and also mention social media – after reading the reference don’t see how’s it all related. Please elaborate or delete or change reference.

Method

4. Line 120 you mention “blind” do you mean VA of NLP both eyes?

5. You told the students about dry eye and then asked them questions about dry eye - Did you have any concerns about that issue? (when you mention something and then ask about it – it might raise awareness)

Results

6. If you had almost 70% Females and DED was more prevalent among females – don’t it change all the statistics? Could you elaborate?

Discussion

7. Thank you for the limitation part.

Conclusion

8. Well written

I think it is an interesting research article. I would consider to publish.

6. PLOS authors have the option to publish the peer review history of their article (what does this mean?). If published, this will include your full peer review and any attached files.

Reviewer #1: **Yes: **Adi Segal

Reviewer #2: No

---

## [Author Response · Author response to Decision Letter 0]

13 Jan 2023

Reviewer 1 Comments 

1. The Literature review sets the stage for the study questions, supported by updated and relevant studies. The literature review is based on relevant studies about self-reported symptoms and clinical examinations of young adults and youth, related to exposure-time to digital media. The authors should re-examine that the review relates to all variables examined, e.g., it lacks information regarding gender differences from previous studies.

In the revised manuscript, the introduction has been reorganized by adding the information regarding the other factors of dry eye, i.e. gender, smoking, contact lens, and perceived stress, as shown in lines 76 – 79. 

2. The abstract points out that smoking and contact lens wear were not predisposing factors for dry eye, however the relevant groups were too small to have a significant statistical conclusion (not mentioned in the limitations section).

Thank you for recommendation. We are truly in agreement with you about the limitation of the small proportion of contact lens wearing and smoking groups. 

Therefore, we added the limitation regarding the small groups, as shown in lines 322 – 325. 

3. The researchers' assumption of dry eye prevalence greater than 50% should be further established, since all the formula calculations are based on this rate. This rate seems high according to the best of my knowledge of the medical literature, especially among younger populations.

We agree with you that this rate may be high among a younger population, but we assumed 50% because:

1) During the first step of the project, we did not have the previous evidence regarding the dry eye symptoms in schoolchildren in the Thai population. 

2) Changes in lifestyle due to the COVID-19 pandemic may prolong screen time in children, which may result in increased dry eye symptoms in children and adolescents. Therefore, we assumed 50% of prevalence to provide the highest number for the sample size. 

Presently, there are several studies that show that the prevalence of dry eye is greater than 50% among university students,1 including Thailand.2 Moreover, the study in China indicated approximately 70% of dry eye among senior secondary school students during the COVID-19 pandemic.3 Thus, this rate is possible in this situation. 

4. The literature review leads to the study aims and the clear research questions but lacks a more specific hypothesis.

In the revised manuscript, this section was re-written and combined with the previous paragraph

as shown in lines 80 – 85. 

5. the sampling process and the research population are well described. The research tools are well described, revealing quite high reliability, although they were originally established for adults. Authors should add information regarding translation validation.

Thank you for your recommendation. In the current version, we have added the information regarding the translation process, as shown in lines 122 – 129. 

6. The authors provided rich information accompanied by detailed tables and plots, demonstrating the higher prevalence among girls, higher grades, prolonged exposure to screens, and higher perceived stress.

The authors add information regarding the relations between dry eye symptoms and perceived stress that was prevalent during the pandemic. In addition, they identified the daily cutoff exposure time to digital media devices, analyzed by odds ratio.

The discussion deals with the findings in relation to previous studies in the literature. The authors demonstrate a critical approach in the study limitations section, especially the lack of clinical evidence to dry eye. It is very relevant in teenagers' populations, which often demonstrate confounding results, especially regarding to self-reports of smoking in the formal educational system context.

We are absolutely agres that the survey in formal educational system is a limitation of smoking factor, so we added this limitation as shown in lines 324 – 325. 

7. Conclusions

The conclusion repeats the study findings and lack the bottom-line importance and significant contribution of the study, as well as lessons learned and recommendations for the future.

Thank you for recommendation. We have revised the conclusion section and provided the significant contribution of the study, as shown in lines 330 – 337. 

Reviwer 2 Comments 

1. You mention “causing a negative impact on ocular surface” but the reference you bring showed only change in OSDI and not the other Ocular surface findings which showed no correlation. 

Thank you for recommendation. We deeply agree that OSDI is not associated with the ocular surface findings. Thus, we have removed this reference. 

We added the study of “Blinking and normal ocular surface in school-aged children and the effect of age and screen time”.4 This study suggested that screen time may be associated with the ocular surface (Tear meniscus height (TMH) and non-invasive tear break-up time (NIBUT)). 

We have also added the study of “Ocular surface predisposing factors for digital display-induced dry eye”.5 This study suggested that digital screen tasks alter the TMH and NIBUT.

2. Line 69-70 you write about increased screen exposure and also mention social media – after reading the reference don’t see how’s it all related. Please elaborate or delete or change reference.

Thank you for recommendation. In the current version, this reference has been deleted.

3. Line 120 you mention “blind” do you mean VA of NLP both eyes?

Yes, we referred to NLP in both eyes. To clarify this, we have revised the manuscript, as shown in lines 117 – 118.

4. You told the students about dry eye and then asked them questions about dry eye - Did you have any concerns about that issue? (when you mention something and then ask about it – it might raise awareness)

Thank you for pointing this out. We are concerned about satisficing (respondents’ answers to the questionnaire are of low quality).6 Most Thai students don’t know about dry eye. Thus, we tried to motivate the guardians and students to respond to the questionnaire by giving them information about dry eye. 

We agree with you that providing the information may raise awareness. However, awareness-raising requires multiple processes for successful implementation.7

5. If you had almost 70% Females and DED was more prevalent among females – don’t it change all the statistics? Could you elaborate?

Thank you for your question. Thus, we stratified the analysis of the female group. The results of significant association are still the same in both the univariate and multivariate analyses.

References 

1. Garcia-Ayuso D, Di Pierdomenico J, Moya-Rodriguez E, Valiente-Soriano FJ, Galindo-Romero C, Sobrado-Calvo P. Assessment of dry eye symptoms among university students during the COVID-19 pandemic. Clin Exp Optom. 2021:1-7.

2. Tangmonkongvoragul C, Chokesuwattanaskul S, Khankaeo C, Punyasevee R, Nakkara L, Moolsan S, et al. Prevalence of symptomatic dry eye disease with associated risk factors among medical students at Chiang Mai University due to increased screen time and stress during COVID-19 pandemic. PLoS One. 2022;17(3):e0265733.

3. Lin F, Cai Y, Fei X, Wang Y, Zhou M, Liu Y. Prevalence of dry eye disease among Chinese high school students during the COVID-19 outbreak. BMC Ophthalmol. 2022;22(1):190.

4. Chidi-Egboka NC, Jalbert I, Wagner P, Golebiowski B. Blinking and normal ocular surface in school-aged children and the effects of age and screen time. Br J Ophthalmol. 2022.

5. Talens-Estarelles C, Garcia-Marques JV, Cervino A, Garcia-Lazaro S. Ocular surface predisposing factors for digital display-induced dry eye. Clin Exp Optom. 2022:1-7.

6. Krosnick JA. Response strategies for coping with the cognitive demands of attitude measures in surveys. 1991;5(3):213-36.

7. Sayers R. Principles of awareness-raising for information literacy: a case study. Bangkok: Communication and Information (CI) UNESCO Asia and Pacific Regional Bureau for Education 2006. 124 p.

---

## [Decision Letter · Decision Letter 1]

20 Feb 2023

PONE-D-22-22320R1Prevalence and associated factors for self-reported symptoms of dry eye among Thai school children during the COVID-19 outbreakPLOS ONE

Dear Dr. Tawonkasiwattanakun,

Thank you for submitting your manuscript to PLOS ONE. After careful consideration, we feel that it has merit but does not fully meet PLOS ONE’s publication criteria as it currently stands. Therefore, we invite you to submit a revised version of the manuscript that addresses the points raised during the review process.

We look forward to receiving your revised manuscript.

Kind regards,

Michael Mimouni

Academic Editor

PLOS ONE

Journal Requirements:

Reviewers' comments:

Reviewer's Responses to Questions

**Comments to the Author**

1. If the authors have adequately addressed your comments raised in a previous round of review and you feel that this manuscript is now acceptable for publication, you may indicate that here to bypass the “Comments to the Author” section, enter your conflict of interest statement in the “Confidential to Editor” section, and submit your "Accept" recommendation.

Reviewer #1: (No Response)

Reviewer #2: All comments have been addressed

2. Is the manuscript technically sound, and do the data support the conclusions?

Reviewer #1: Yes

Reviewer #2: Yes

3. Has the statistical analysis been performed appropriately and rigorously? 

Reviewer #1: I Don't Know

Reviewer #2: Yes

4. Have the authors made all data underlying the findings in their manuscript fully available?

Reviewer #1: Yes

Reviewer #2: Yes

5. Is the manuscript presented in an intelligible fashion and written in standard English?

Reviewer #1: Yes

Reviewer #2: Yes

6. Review Comments to the Author

Reviewer #1: The authors of the manuscript have definitely shown improvement in the structure, and the improvement is noticeable throughout the entire manuscript. The authors have addressed all previous comments. The sequence of the paper is better organized. The literature review is more robust and up to date. The method and results sections are much clearer and transparent. The authors use the added references in the discussion, making it easier for the reader to understand how the work they have done contributes to current scientific literature.

I now find the manuscript publishable, however I suggest four additional minor revisions.

• Line 71: The authors mentioned the importance of TFOS epidemiologic report. I recommend adding a few words to summarize its main conclusions rather than its call for additional research.

• Line 117: Lack of light perception, the precise professional definition is "No light perception".

• Line 254, should read: There are no validated questionnaires for "dry eye symptoms", or cut off scores…

• The paragraph starting from 277: The authors added important comparison with other dry eye studies which used different questionnaires. The reader can benefit from some more critical information regarding the different or similar parameters those questionnaires used.

Reviewer #2: Thanks for reading, understanding and integrating our comments.

I think this new paper much better and ready for publication.

7. PLOS authors have the option to publish the peer review history of their article (what does this mean?). If published, this will include your full peer review and any attached files.

Reviewer #1: No

Reviewer #2: **Yes: **Evgeny Gelman

---

## [Author Response · Author response to Decision Letter 1]

4 Mar 2023

Reviewer #1: 

General comment: The authors of the manuscript have definitely shown improvement in the structure, and the improvement is noticeable throughout the entire manuscript. The authors have addressed all previous comments. The sequence of the paper is better organized. The literature review is more robust and up to date. The method and results sections are much clearer and transparent. The authors use the added references in the discussion, making it easier for the reader to understand how the work they have done contributes to current scientific literature.

I now find the manuscript publishable, however I suggest four additional minor revisions.

1. Line 71: The authors mentioned the importance of TFOS epidemiologic report. I recommend adding a few words to summarize its main conclusions rather than its call for additional research.

Thank you for you recommendation. We revised this statement and added the summary from the TFOS epidemiology report (line 72-74).

2. Line 117: Lack of light perception, the precise professional definition is "No light perception".

Thank you for your suggestion. In revised manuscript, we used “no light perception” (line 118).

3. Line 254, should read: There are no validated questionnaires for "dry eye symptoms", or cut off scores. 

Thank you for pointing this out; we revised this statement (line 254)

4. The paragraph starting from 277: The authors added important comparison with other dry eye studies which used different questionnaires. The reader can benefit from some more critical information regarding the different or similar parameters those questionnaires used.

Thank you for your suggestion. We revised this paragraph and provided details of these questionnaires. We believe that the reader will benefit from this section (in lines 280-294).

Reviewer #2: 

Thanks for reading, understanding and integrating our comments. I think this new paper much better and ready for publication.

We appreciate your kind comment and thank you for reviewing our manuscript.

---

## [Decision Letter · Decision Letter 2]

12 Apr 2023

Prevalence and associated factors for self-reported symptoms of dry eye among Thai school children during the COVID-19 outbreak

PONE-D-22-22320R2

Dear Dr. Tawonkasiwattanakun,

We’re pleased to inform you that your manuscript has been judged scientifically suitable for publication and will be formally accepted for publication once it meets all outstanding technical requirements.

Kind regards,

Michael Mimouni

Academic Editor

PLOS ONE

Additional Editor Comments (optional):

Reviewers' comments:

Reviewer's Responses to Questions

**Comments to the Author**

1. If the authors have adequately addressed your comments raised in a previous round of review and you feel that this manuscript is now acceptable for publication, you may indicate that here to bypass the “Comments to the Author” section, enter your conflict of interest statement in the “Confidential to Editor” section, and submit your "Accept" recommendation.

Reviewer #1: All comments have been addressed

Reviewer #2: All comments have been addressed

2. Is the manuscript technically sound, and do the data support the conclusions?

Reviewer #1: Yes

Reviewer #2: Yes

3. Has the statistical analysis been performed appropriately and rigorously? 

Reviewer #1: I Don't Know

Reviewer #2: Yes

4. Have the authors made all data underlying the findings in their manuscript fully available?

Reviewer #1: Yes

Reviewer #2: Yes

5. Is the manuscript presented in an intelligible fashion and written in standard English?

Reviewer #1: Yes

Reviewer #2: Yes

6. Review Comments to the Author

Reviewer #1: The authors have addressed all my comments. The claims are coherent, and the analysis is simple and clear. I now find the manuscript publishable.

Reviewer #2: Thanks for reading, understanding and integrating our comments.

I think this new paper much better and ready for publication.

7. PLOS authors have the option to publish the peer review history of their article (what does this mean?). If published, this will include your full peer review and any attached files.

Reviewer #1: **Yes: **Adi Segal

Reviewer #2: No

---

## [Editor Report · Acceptance letter]

16 Apr 2023

PONE-D-22-22320R2 

Prevalence and associated factors for self-reported symptoms of dry eye among Thai school children during the COVID-19 outbreak 

Dear Dr. Tawonkasiwattanakun:

I'm pleased to inform you that your manuscript has been deemed suitable for publication in PLOS ONE. Congratulations! Your manuscript is now with our production department. 

Kind regards, 

on behalf of

Dr. Michael Mimouni 

Academic Editor

PLOS ONE